# A Deep Dive into the N-Terminus of STIM Proteins: Structure–Function Analysis and Evolutionary Significance of the Functional Domains

**DOI:** 10.3390/biom14101200

**Published:** 2024-09-24

**Authors:** Sasirekha Narayanasamy, Hwei Ling Ong, Indu S. Ambudkar

**Affiliations:** Secretory Physiology Section, National Institute of Dental and Craniofacial Research, National Institutes of Health, Bethesda, MD 20892-1190, USA; sasirekha.narayanasamy@nih.gov (S.N.); ongh@mail.nih.gov (H.L.O.)

**Keywords:** EF-hand, SAM, STIM1, STIM2, SOCE, calcium signaling

## Abstract

Calcium is an important second messenger that is involved in almost all cellular processes. Disruptions in the regulation of intracellular Ca^2+^ levels ([Ca^2+^]_i_) adversely impact normal physiological function and can contribute to various diseased conditions. STIM and Orai proteins play important roles in maintaining [Ca^2+^]_i_ through store-operated Ca^2+^ entry (SOCE), with STIM being the primary regulatory protein that governs the function of Orai channels. STIM1 and STIM2 are single-pass ER-transmembrane proteins with their N- and C-termini located in the ER lumen and cytoplasm, respectively. The N-terminal EF-SAM domain of STIMs senses [Ca^2+^]_ER_ changes, while the C-terminus mediates clustering in ER-PM junctions and gating of Orai1. ER-Ca^2+^ store depletion triggers activation of the STIM proteins, which involves their multimerization and clustering in ER-PM junctions, where they recruit and activate Orai1 channels. In this review, we will discuss the structure, organization, and function of EF-hand motifs and the SAM domain of STIM proteins in relation to those of other eukaryotic proteins.

## 1. Introduction

Calcium (Ca^2+^) is a versatile intracellular messenger that plays a crucial role in a myriad of cellular processes such as fertilization, cell differentiation, muscle contraction, cell motility, cellular metabolism, synaptic plasticity, RNA transcription, and secretion [1,2,3,4,5,6,7,8]. Abnormal Ca^2+^ signaling leads to severe pathological conditions such as ischemia, stroke, and autoimmune diseases, as well as immune deficiency. It has also been implicated in neurological disorders such as Alzheimer’s, Huntington’s, and Parkinson’s disease [9,10,11]. The mechanisms of intracellular Ca^2+^ signaling are diverse in different cell types, involving a number of molecular signaling components such as receptors, pumps, and effector molecules that are organized into functionally distinct combinations to mediate Ca^2+^ signaling as required for cell function [2]. Specific Ca^2+^ sensor proteins can transduce the initial Ca^2+^ signal into the regulation of multiple cellular functions. For example, local Ca^2+^ near the Orai1 channel is sensed by calmodulin (CaM) and utilized to regulate a variety of signaling processes, such as NFAT-dependent gene expression. On the other hand, global Ca^2+^ signals can be sensed by a different set of Ca^2+^ sensors and used to regulate other Ca^2+^-dependent cell functions. The magnitude and frequency of the Ca^2+^ signal, as well as its spatial organization, play key roles in the type of Ca^2+^ sensor involved.

As such, cells have several mechanisms to regulate cytosolic [Ca^2+^]_i_: (i) plasma membrane calcium channels mediate extracellular Ca^2+^ influx into the cytosol; (ii) sequestration of cytosolic Ca^2+^ into the intracellular organelles, such as the ER and mitochondria, via Ca^2+^ pumps and channels; (iii) extrusion of Ca^2+^ from the cytoplasm to the extracellular milieu via plasma membrane Ca^2+^ pumps and exchangers [12]. In unstimulated or resting cells, [Ca^2+^]_i_ is maintained at a concentration between 10 and 100 nM. Following cell stimulation, [Ca^2+^]_i_ can increase several folds depending on the stimulus intensity and type of cells [13]. Cells are equipped with the ability to sense and respond accordingly to physiologically low and relatively high stimulus intensities [12,14]. The Ca^2+^-binding proteins (CaBPs) function as Ca^2+^ buffers and sensors that respond to changes in the Ca^2+^ levels and activate a cascade of downstream effector molecules to transduce the signal into specific responses and prevent unregulated increases in [Ca^2+^]_i_. The Ca^2+^-binding affinities in these proteins are determined by specific motifs and domains [15,16].

Store-operated calcium entry (SOCE) is a ubiquitously present Ca^2+^ entry mechanism that regulates a variety of physiological functions [17,18]. SOCE was first identified by Putney, wherein depletion of ER-Ca^2+^ stores results in the activation of plasma membrane Ca^2+^ channels [19]. The cascade of steps that lead to SOCE starts with the stimulation of G-protein coupled receptors (GPCRs), activation of phospholipase C (PLC), and conversion of phosphatidylinositol-4,5-bisphosphate (PIP_2_) into diacylglycerol (DAG) and inositol trisphosphate (IP_3_). IP_3_ binds to the IP_3_ receptor (IP_3_R) present on the ER membrane and elicits Ca^2+^ release from the ER into the cytoplasm, a process known as ER-Ca^2+^ store depletion [20,21]. While ryanodine receptors can contribute to IP_3_-induced Ca^2+^ store depletion via Ca^2+^-induced Ca^2+^ release or via IP_3_-independent mechanisms, this review focuses on the IP_3_R-mediated mechanism. SOCE is primarily dependent on the Ca^2+^-selective plasma membrane channel family, Orai, comprising Orais1, 2, and 3, and multi-domain Ca^2+^-binding proteins, stromal interacting molecules STIM1 and STIM2, which are single-pass transmembrane (TM) proteins in the ER membrane, with N-terminal EF-hands in the ER lumen that sense the decrease in [Ca^2+^]_ER_. STIM1 and Orai1 form the elementary units for SOCE, with the Orai1 channel mediating the Ca^2+^ selective Ca^2+^ current (I_CRAC_). In unstimulated or resting cells, STIM1 is diffused throughout the ER and also tracks along the microtubules by interacting with the EB1 protein. Orai1 is freely diffused in the plasma membrane. Following ER-Ca^2+^ store depletion, STIM1 loses the Ca^2+^ bound to its canonical EF (cEF)-hand, which triggers a conformational change both in its N and C termini, causing the protein to dimerize and translocate to the ER-plasma membrane junctions (ER-PM junctions). Orai1 is recruited into these junctions and gated via interactions with STIM1 [22,23,24]. On the other hand, STIM2 is already pre-clustered in the ER-PM junctions in unstimulated/resting cells, which further increases following ER-Ca^2+^ store depletion. STIM2 also recruits and activates Orai1, albeit with lower efficiency and at different stimulus intensities when compared to STIM1 [17,25,26] (Figure 1).

There is considerable interest in elucidating the structure-to-function relationship of STIM and Orai proteins, judging from the large number of publications in the past two decades. This review will discuss the N-terminal domain of the STIM proteins, elaborating on the structural organization and evolutionary significance of the EF-hand and SAM domains. Even though both STIMs can also interact with Orai2 and Orai3, this review will only focus on STIM–Orai1 interactions while briefly describing the evolution of Orai channels.

## 2. Components of Store-Operated Calcium Entry (SOCE)

Orai1 was discovered in an RNAi-based screen of SOCE components in *Drosophila* S2 cells and linkage analysis of patients with severe combined immunodeficiency [29,30,31,32]. Almost all invertebrates, such as nematodes, fruit flies, mosquitoes, sea urchins, and seq squirts, have a single Orai protein. This evolved into Orai1 and Orai2 in vertebrates, as shown for fish and frogs. Orai3 is formed via duplication of the *Orai1* gene and is exclusive to mammals [29,33,34]. Unlike Orai1, the knockdown of Orais 2 and 3 did not abrogate SOCE and I_CRAC_. Nonetheless, both Orais contribute to other (non-SOCE) types of Ca^2+^ influx [34]. All three Orais show near complete conservation within the pore-lining TM1 and highly identical TMs 2 to 4 but diverging cytosolic N- and C-termini sequences. There are 34% and 46% sequence similarity in the N- and C-termini of Orais 1 and 3. Additionally, Orai3 exhibits a longer third extracellular loop than Orais 1 and 2 [33,34,35]. Orai1 has two isoforms, Orai1α (longer, approximately 33 kDa) and Orai1β (lacks 63 aa in the N-terminus). Orai1β arose from an alternative translation initiation site in Orai1α and did not contain a poly-arginine sequence suggested to mediate Orai1 interaction with plasma membrane PIP_2_. As such, Orai1β exhibited faster mobility than Orai1α, but it was unclear how that would impact channel function since both isoforms generated similar magnitudes of SOCE and I_CRAC_. Interestingly, Orai1α and Orai1β form only homomeric CRAC channels [33,34,36]. In a study using a weaker TK promoter to drive Orai1 expression in mouse embryonic fibroblasts from Orai1^−/−^ mice (Orai1-KO MEFs), both isoforms restored I_CRAC_ in Orai1-KO MEFs to similar levels in wild-type MEFs. Orai1α-mediated I_CRAC_ exhibits a faster CDI as rapid inactivation was observed with EGTA (slower chelator) but not BAPTA (fast chelator). However, no difference was observed for Orai1β-mediated I_CRAC_. Both Orai1α and Orai1β supported I_SOC_ when co-expressed with TRPC1 and STIM1, but only Orai1α contributed to I_ARC_ (arachidonic acid-induced current). The authors proposed that mammalian evolution exploited the process of alternative translation start sites to generate different isoforms of the same channel with distinct modes of regulation and contribution to cellular processes [33,34,37].

Orai arose earlier than STIM during evolution and may have initially contributed to non-IP_3_-mediated calcium influx. When STIM arose as a complement to IP_3_/IP_3_R signaling, the existing Orai channels may have been co-opted for SOCE [12,38,39,40]. The key residue controlling Ca^2+^ selectivity (E106 in Orai1) is conserved in many species, suggesting that Orai proteins have long functioned as highly Ca^2+^-selective channels [12]. TBlastN searches using *Caenorhabditis elegans* Orai protein revealed a putative cation transporter (PfuCaT) from the genomic sequence of *Pyrococcus furiosus*, with homologs of PfuCaT also present in *Pyrococcus horikoshii* and *Thermococcus kodakaerensis*. All three cocci are hyperthermophilic archaea found in isolated environments with high temperatures. While Orai has four TMs, PfuCaT has five, with TM1 linked by a large hydrophilic loop to the rest. Nonetheless, PfuCaT shares a highly conserved signature motif with two internal repeats containing two key residues essential for Ca^2+^ selectivity (similar to E106 and E190 in Orai1). Since there is limited sequence similarity outside of the signature motif, PfuCaT is likely to have different regulatory mechanisms than Orai. Notably, sequences similar to the coiled-coil ezrin–radixin–moesin (ERM) domain of STIM1 have been found in hyperthermophilic archaea genomes, suggesting that both Orai and STIM proteins may have adapted existing domains to support SOCE during the transition to metazoans [29,41]. While Orai proteins in nematodes and insects have a long N-terminus and short TM3-TM4 loop, those in ascidians, non-mammal vertebrates, and mammals have a short N-terminus but longer TM3-TM4 loop. For example, the N-termini of *C. elegans* and *Drosophila melanogaster* Orai1 have 119 aa and 160 aa c.f. 66 aa in *Ciona intestinalis* (sea squirt). Orai3, which is mammalian-specific, has the longest TM3-TM4 loop (72 aa) c.f. *C. elegans* (13 aa) and *D. melanogaster* (9 aa). The author proposed that changes in the lengths of the N-terminus and TM3-TM4 loop reflect functional evolutionary changes between species [29].

STIM proteins were discovered by two independent groups using RNAi screens to identify SOCE components. Roos et al. identified the role of STIM in SOCE by knocking down the expression of STIM in lower eukaryotes, such as in fruit flies and *D. melanogaster* [42]. The second group used gene silencing techniques to identify both homologs of STIM (STIMs 1 and 2) that are involved in sensing ER-Ca^2+^ store depletion and SOCE in HeLa cells [43]. In higher organisms such as vertebrates, starting from fishes, STIM proteins have evolved into two types: STIM1 and STIM2 [12]. While gene localization studies indicate that STIM1 and STIM2 are localized in chromosomes 11 and 4, respectively, both proteins share high sequence and structural similarity. The functional regions of vertebrate STIM consist of identical domains such as the N-terminal EF-SAM domain in ER lumen, a single-pass ER-TM, followed by cytosolic coiled-coil domains, a STIM-Orai activating region (SOAR), a Serine/Proline-rich (in STIM1) or Proline/Histidine-rich (in STIM2) domain, and a polybasic domain in the C-terminus. Nonetheless, these homologs have evolved to perform different functions. STIM1 has a high Ca^2+^-binding affinity endowed by its canonical EF (cEF) hand and requires substantial ER-Ca^2+^ store depletion to be activated. Conversely, the cEF-hand of STIM2 has a lower affinity that enables the protein to sense smaller levels of ER-Ca^2+^ store depletion. The sensing of sub-threshold stimuli by STIM2-cEF-hand results in the immobilization of STIM2 clusters within the ER-PM junctions in resting cells. As such, both proteins behave differently in resting cells where STIM2 is already pre-clustered in the ER-PM junctions but not STIM1. Moreover, the SOAR of STIM2 is a weaker activator of Orai1 when compared to STIM1. STIM2 contributes to constitutive or basal Ca^2+^ entry in resting cells by interacting with and gating Orai1. Notably, pre-clustered STIM2 can recruit STIM1 through the C-terminal coiled-coil domain interaction in the cytoplasm and enhance STIM1/Orai1 coupling in cells with relatively high [Ca^2+^] in the ER lumen ([Ca^2+^]_ER_), such as in resting cells or when cells are stimulated with relatively low [agonist] [26,44,45,46,47]. Therefore, the N-terminal region of STIM proteins plays an important role in transducing stimuli into SOCE activation. The three-dimensional structures reported for human STIM1 (PDB: 2K60) and STIM2 (PDB: 2L5Y) contain a canonical and a non-canonical EF-hand motif followed by the sterile-α-motif (SAM) [27,28]. In the following sections, we will discuss the structural organization of EF-hand motifs and SAM domains present in STIMs in relation to other CaBPs and multi-domain-containing eukaryotic proteins.

## 3. EF-Hand Motifs in Calcium-Binding Proteins: Overview and Properties

The EF-hand motif was first identified by Kretsinger while determining the structure of parvalbumin [48]. This motif is ~30 amino acids in length and folds into two α-helices (entering (E) and exiting (F) helices), which are almost perpendicular to each other in Ca^2+^-bound form. These helices are connected by a 12-residue Ca^2+^-binding loop that coordinates Ca^2+^ by attaining a pentagonal bipyramidal geometry [48]. This motif has been identified in several CaBPs, including ELC, RLC, troponin C, and CaM. About 156 subfamilies of EF-hand motifs and 3000 EF-hand-related entries can be found in the NCBI Reference Sequence Database [49,50]. Intriguingly, some CaBPs encompass only EF-hand motifs (e.g., CaM [51], troponin C [52], calumenin [53]), while in other proteins, EF-hand motifs are part of a large multi-domain protein. In those latter proteins, EF-hand motifs are found either at the N-terminal, middle, or C-terminal region of the protein [49,54]. While EF-hands are associated with the SAM domain in STIM proteins, other EF-CaBPs have cysteine protease, leucine zipper domain, and coiled coil domain associated with it [27,28,55,56,57]. Two adjacent EF-hand motifs organized into a four-helix bundle domain and occurring in pairs are known as the EF-hand domain, which is observed in many proteins, including troponin C [52,58], calbindin D_9k_ [59], and SCP [60]. A central hydrophobic core formed by domain packing of the E and F helices drives the dimerization of EF-hand motifs [61,62]. An anti-parallel β-sheet established between the EF-hand loop sequences additionally stabilizes its structural integrity, a common property that is present in many EF-hand proteins [50,63]. The EF-hand domains are arranged into different spatial conformations that primarily depend on the length of a linker region connecting the two EF domains. For example, the four Ca^2+^-bound EF-hand motifs of calmodulin (CaM) are arranged into two N and C-terminal EF-hand globular domains connected by a linker [51,64]. The essential function of the EF-hand motifs is to translate the changes in the local Ca^2+^ concentration into a physiological response. EF-hand motifs exhibit diversity in order to achieve versatility with regards to structure, Ca^2+^-binding, conformations, and domain organizations [61,65]. These motifs bind Ca^2+^ with dissociation constants (K_D_) ranging from 10^−9^ to 10^−4^ M. The Ca^2+^ binding affinities vary among the EF-hand proteins and primarily arise from different amino acids found in the loop region and their intermolecular interactions with target proteins [63]. The EF-hand motifs possess diverse metal ion bind properties and can bind other similar-sized cations, such as Mg^2+^, to form a Ca^2+^/Mg^2+^ mixed site [66,67,68]. However, EF-hand motifs adopt different geometries accordingly. Whereas Ca^2+^ is bound by seven ligands in the EF-hand loop arranged in a pentagonal bipyramidal configuration, Mg^2+^ utilizes only six ligands to form an octahedral coordination scheme [63,69].

### 3.1. Structural Response to Ca^2+^-Binding and the Stability

EF-hand motifs undergo conformational/structural changes upon coordinating Ca^2+^, which is then recognized by target effector molecules to relay the Ca^2+^ signals. These Ca^2+^-induced structural changes have been well documented for several CaBPs using biophysical techniques, molecular dynamics simulation, and other structural characterization methods. Based on the Ca^2+^-induced conformational changes, they are classified into two classes: Ca^2+^ sensors and Ca^2+^ buffers. The key property of Ca^2+^ sensors is their ability to undergo major conformational changes upon binding Ca^2+^ and present an open conformation with their hydrophobic pocket exposed to the solvent, which serves as a target protein recognition and interaction site [50]. Ca^2+^ binding-induced transition from a closed to an open conformation has been reported for CaM [70], troponin C [71], and S100 family members [72]. Ca^2+^ buffers are a small population of cytosolic Ca^2+^-binding proteins that undergo very little/no conformational changes upon Ca^2+^-binding (e.g., calbindin (D_9k_, D_28k_), α and β-parvalbumin, and calretinin [73,74]) and regulate short-lived Ca^2+^ signals by modulating their spatial–temporal aspects [50,75]. The ability of these types of Ca^2+^- buffers to modulate intracellular Ca^2+^ signaling depends on several factors, such as Ca^2+^ concentration, affinity to bind Ca^2+^ (and other similar metal ions), Ca^2+^-binding kinetics, and mobility of both Ca^2+^-buffering proteins and the Ca^2+^ ions [75].

### 3.2. Diversity of the Amino Acids in the Loop Region

The loop region that connects the E and F helices of an EF-hand motif contains 12 amino acid residues that are arranged in a pentagonal bipyramidal fashion to coordinate a single Ca^2+^ ion. The assignment of the amino acids in a typical EF-hand motif is as follows: 1(+X), 3(+Y), 5(+Z), 7(−Y), 9(−X), and 12(−Z), where the numbers indicate the position of amino acids in the loop that contribute oxygen (O) atoms to coordinate Ca^2+^ and the axes within brackets indicate their tertiary geometry in a pentagonal bipyramidal configuration. The Ca-O bond stabilizes this interaction, wherein the oxygen is provided by the main-chain carbonyl oxygen atoms, side chains of the following amino acids such as Asp, Asn, Glu, Gln, and Ser, and a water molecule [62,76]. Therefore, the loop regions of EF-hand motifs are rich in polar and negatively charged amino acids. Of note, a web of hydrogen bonds stabilizes the Ca^2+^ coordination sphere in addition to the Ca-O bond and diminishes the electrostatic repulsion of the negatively charged amino acids in the loop region [63]. The EF-loops of the Ca-binding proteins are divided into two types based on length, amino acid composition, and tertiary geometry, namely canonical and non-canonical loops. The canonical loop in an EF-hand motif contains 12-mer that contributes to protein stabilization and helps to maintain the overall domain fold. The importance of amino acid residues found at each position is discussed below. The first (X) and third (Y) positions in an EF-loop participate in the stereochemical arrangement of the loop and are occupied by Asp at a relatively higher frequency than any other amino acids. Similarly, Asp occupies the 5th (Z) position in an EF loop. The 6th position in an EF-hand loop usually contains Gly (96%), which provides an uncommon main chain conformation to bend the loop at an angle of ~90° that enables the remaining Ca^2+^ ligands to occupy the coordinating position [50,63,76]. Nonetheless, amino acids other than Gly have been reported in EF-hand motifs of polycystin-2 (PC-2) and calumenin at the 6th position, wherein the presence of amino acids other than Gly has been found to have a dramatic effect on the Ca^2+^-binding affinity and protein folding [77,78]. The following hydrophobic amino acids, Ile, Val, and Leu, occupy the 8th position in the loop, which enables the formation of a short anti-parallel β-sheet that is crucial for dimerization of the EF-hand motifs to form an EF-hand domain. The 9th position (−X) is preferably occupied by Asp, which chelates the Ca^2+^ ion through hydrogen bonding or a bridging water group. In addition, the amino acid at the 9th position initiates the F helix, which is stabilized by additional hydrogen bonding provided by the side chain of the Glu at the 12th position (−Z) [50,62,76]. Hence, the amino acids present in the EF-loop regions are of key importance in not only chelating metal ions but also stabilizing the structure of the EF-hand motif. 

Despite this, several EF hands have been reported to coordinate Ca^2+^ through an uncommon mechanism and are called non-canonical EF (nEF) hands. These nEF-hand loops vary in amino acid composition and length, contributing to the functional diversity of EF-hands [63]. For example, nEF-hand motif 3 (EF3) of calcium and integrin binding protein (CIB) possesses a similar length in the loop region but with Asp instead of Glu at the 12th position. This results in a shorter EF3 loop and has an impact on the Ca^2+^ coordination sphere as well as ion selectivity [79,80,81]. In some nEF-hands, the Ca^2+^ coordination loop is longer than a typical cEF-hand due to the insertion of additional amino acids (e.g., EF5 of ALG-2). In such EF-loops, Ca^2+^ is chelated only by the N-terminus of the loop and by an additional water molecule [82]. Interestingly, some of the nEF-hands are shorter than the normal length and contain only eleven residues within the loop. Examples of this type are observed in the EF-hand of calpain domain VI, where the absence of the canonical 1st and 3rd Ca^2+^-ligands is compensated by oxygens contributed by the main chain carboxyl groups and water molecules [83].

### 3.3. Organization of EF-Hands in STIM Proteins 

The N-terminal region of STIM proteins, consisting of nEF and cEF motifs as well as the SAM domain, share high sequence homology and similar domain architecture. Studies have reported that the N-terminal of STIM1 plays a crucial role in SOCE, as mutations in the EF-hands or SAM show major differences in physiological function [18]. Purified recombinant EF-SAM region of STIM1 elutes as a dimer or oligomer in the absence of Ca^2+^. However, in the presence of Ca^2+^, the protein elutes in a monomeric form. Consistent with this, secondary structure determination using far-UV CD spectroscopy indicates that the EF-SAM exists in a random coil-like conformation, and STIM1 adopts an α-helical structure upon binding Ca^2+^ [27,84]. This Ca^2+^-induced structural transition is a signature of many EF-hand-containing Ca^2+^-sensor proteins, where the Ca^2+^-induced structural transitions play an important role in protein–protein interaction and target recognition [85,86]. In addition, Ca^2+^-bound STIM1 (holo-STIM1) is more thermostable than Ca^2+^-free STIM1 (apo-STIM1), indicating that Ca^2+^-binding imparts structural stability. Furthermore, STIM1 binds Ca^2+^ with a relatively high affinity (K_D_ ~200–600 mM) when compared to STIM2 [84,87]. Recent studies have shown that the GrpE-fused EF-SAM domain of STIM1 can bind ~5 Ca^2+^ ions/monomer as determined by isothermal titration calorimetry studies [88]. Consistent with this, molecular dynamics simulations showed the clustering of Ca^2+^ ions around the nEF-hand and SAM domain of STIM1, in addition to cEF-hand [89]. Interestingly, the binding of Ca^2+^ at these additional sites is solely dependent on the cEF-hand’s Ca^2+^-binding ability, as it is not detected in the case of the GrpE-fused D76A-EF-SAM mutant of STIM1 (which lacks the ability to bind Ca^2+^) [88]. Although STIM1 contains an EF-hand pair, the NMR structure indicates that it coordinates only one Ca^2+^ at the cEF-hand [87]. The nEF hand lacks the ability to bind Ca^2+,^ which might be due to the presence of unconventional amino acids in its loop region (1st, 3rd, 6th, and 9th positions).

Atomic-level structural information available for the Ca^2+^-bound EF-SAM domain of STIM1 (PDB: 2K60) sheds light on the molecular details of the STIM1 [27]. The N-terminal region of EF-SAM is organized into a single globular domain containing predominantly α-helical secondary structural elements. The NMR structure shows a canonical helix–loop–helix Ca^2+^-binding EF-hand, a non-canonical EF-hand motif (also known as hidden EF-hand), and a SAM domain. Like other EF-hand pair-containing proteins, the cEF-hand of STIM1 also pairs with the nEF-hand. Of note, the NMR structure of STIM1 indicates that the second nEF-hand is stabilizing the cEF-hand via hydrogen bonding between the respective EF-loop regions (C(O) Val83, Ile115: N(H) Val 83, Ile115) and forming a small anti-parallel β-sheet, a property reminiscent of several EF-hand containing CaBPs [27,63,90]. In addition, Stathopulos et al. have reported that the STIM1 EF-hand pair shows high homology (structural alignment) to the C-terminal of Ca^2+^-bound bovine CaM. Also, vector-geometry mapping analysis measuring the interhelical angles between the E and F helices indicates that STIM1 EF-hands exist in an open conformation similar to that observed in Ca^2+^-bound CaM structures [27]. The open conformation of Ca^2+^-bound STIM1 may expose hydrophobic surfaces that pack against the non-polar surfaces of the SAM domain. However, as the apo-STIM1 structure has not been described yet, it is difficult to delineate how STIM1 structurally transitions from its apo to holo form. 

To understand the evolutionary significance of EF-hands among the STIM1 orthologs, we performed multiple sequence alignment of STIM1 proteins from different species (Figure 2). Briefly, human STIM1 (Uniprotkb ID: Q13586) was used as a reference sequence and blasted against non-redundant sequences in the NCBI BLAST (protein blast)/Uniprotkb (Swissprot) server [91,92]. About ~154 STIM1 protein sequences were collected from different animals, and the sequences were checked for redundancy. Since STIM1 protein exists in different isoforms, only the longest isoform from each animal was used for multiple sequence analysis. We compared protein sequences from different animal species, such as monkeys, tortoises, turtles, mole rats, bats, squirrels, and lemurs, for similarities/differences. Only one unique representative protein sequence from each genus of the above-mentioned animals was used in the alignment. About ~70 STIM1 orthologous protein sequences were aligned using the MAFFT multiple sequence alignment tool [93]. MSA shows that STIM1 is an evolutionarily conserved protein among the metazoan tree of the family and is absent in plants, fungi, and bacteria. In the lower metazoans such as Arthropoda (*D. melanogaster*) and Nematoda (*C. elegans*, roundworm, etc.), STIM exists as a single copy [12,94]. Previous studies have shown that the STIM family might have undergone two rounds of gene duplication in vertebrates, one as early in the euteleostomi lineage and another within the fish genome [94], to give rise to STIM1 and STIM2 that perform different physiological functions. In the higher metazoans, such as in higher vertebrates, STIM1 is an evolutionarily conserved protein and can be found in different classes of Chordata phyla (i.e., mammals, birds, reptiles, amphibians, and fish). All STIM orthologs found in the Chordata phyla contain both the canonical and non-canonical EF hands. Comparing the 12-amino acid sequences of the cEF-hand loop of human STIM1 with other animals within the Chordata phyla, it is evident that the Ca^2+^ coordinating amino acids are 100% conserved except for the amino acid at the 7th position, wherein Asp (D) is replaced by Asn (N) in reptiles and birds and Ser (S) in zebrafish. On the other hand, STIM of the lower metazoans, such as *D. melanogaster* and nematodes, have Asn (N) and Ser (S), respectively, at that position. As few variabilities in the amino acids have been observed at the 4th, 10th, and 11th positions, the Ca^2+^-binding capacity of STIM1 is well-preserved during the evolution from lower organisms to higher metazoans.

It is hypothesized that Ca^2+^ binding to the nEF hand might have been lost during evolution. In our multiple sequence analysis (Figure 2), we found that the unusual amino acids in the 1st, 3rd, 6th, and 9th positions of the Ca^2+^-binding loop region are highly conserved in higher vertebrates. While few variations have been found in lower metazoans, such as in *D. melanogaster, C. elegans,* and worms, these are not equivalent to the amino acids observed in a typical Ca^2+^-binding cEF-hand such as in CaM or other canonical EF-hand motifs. Therefore, STIM1 likely evolved to accommodate only one functional Ca^2+^-binding EF-hand motif. Despite lacking the Ca^2+^ binding site, the nEF-hand motif might play an important role in maintaining the integrity of the EF-hand pair by pairing with the cEF-hand [27,88,89]. These observations indicate that the EF hands of STIM1 play an important role in Ca^2+^ sensing by the protein and its regulation of SOCE.

Although the EF-SAM domain of STIM2 possesses a high sequence identity (~58%) to STIM1, it differs in various biophysical and biochemical properties. In contrast to apo-STIM1, STIM2 is a natively folded protein and exhibits an α-helical secondary structural element in the absence of Ca^2+^, as determined by far-UV CD spectroscopy. Thermal denaturation studies indicate that apo-STIM2 retains its secondary structure and exhibits a higher melting temperature (T_m_) when compared to apo-STIM1 [84]. Further addition of Ca^2+^ leads to minor changes in the CD spectra of STIM2, akin to the properties of several Ca^2+^-buffering proteins [75]. Although STIM2 does not undergo major conformational changes upon Ca^2+^-binding (as observed for STIM1), Ca^2+^-induced conformational spectral shifts were observed [84]. In addition, apo-STIM2 elutes at a slightly higher elution volume than holo-STIM2. These results indicate that apo-STIM2 exists in a slightly extended form, whereas holo-STIM2 is compact and well-folded. Furthermore, STIM2 binds Ca^2+^ with apparently low-binding affinity (K_D_ of ~0.5–0.8 mM [28,44,84]) (c.f. K_D_ of ~0.2–0.6 mM for STIM1 [84,87]).

The NMR structure of STIM2 EF-SAM (PDB: 2L5Y) reveals many interesting details about the domain organization of STIM2 [28]. Similar to STIM1, STIM2 also contains a Ca^2+^-binding cEF hand, a nEF hand, and a SAM domain. Despite lacking the Ca^2+^-binding site, the nEF-loop stabilizes the cEF-loop through hydrogen bonding between N(H) of Ile87 and C(O) of Ile119 to form EF-hand pairs. The tertiary structure of STIM2-EF-SAM is negatively charged at pH 7, which is mainly contributed by the amino acids present in the EF-hand pair. Like STIM1, the cEF-hand of STIM2 exists in an open conformation, which is measured by the interhelical angles, which are about > 80°. Because of this, STIM2 exposes large hydrophobic surfaces contributed by the following amino acids in the cEF-hand, namely Leu72, Ile75, His76, Met79, and Phe85. The amino acids from the nEF hand, such as Met100, Lys103, Lys108, Leu112, Ile119, Leu124, and Trp128, equally contribute to the hydrophobic patches. Together, the contribution of hydrophobic surfaces from the EF-hand motifs serves as a dock for its interaction with the hydrophobic regions of SAM [28].

## 4. Sterile Alpha Motif (SAM)

Sterile alpha motif, also known as SAM, is a well-conserved domain (~70 amino acids in length) that was first discovered in yeast proteins involved in differentiation from haploid to diploid cells [95,96,97]. It was later identified in many proteins from different kingdoms, such as in eukaryotes, bacteria, and viruses. Within the eukaryotes, SAM proteins have been found in mammals, such as humans, chimpanzees, rats, and mice, as well as in insects, fishes, frogs, worms, and birds. Recent SMART non-redundant database searches show that there are 64,647 SAM domains seen in 49,933 proteins, indicating that the SAM domain exists either as a single unit or in tandem repeats within a protein [98]. Interestingly, SAM domains are largely seen in multi-domain-containing proteins localized to different cellular compartments, indicating their diversified and extensive cellular functions [96].

The structural organization of the SAM domain consists of a five α-helix bundle where the α3 helix is shorter in length than the rest of the helices. SAM domains interact homomerically with each other or through heteromeric interactions with other proteins. Interestingly, both the homo and heterotypic interactions are guided through head-to-tail interactions, well known as the Mid-Loop (ML)/End-Helix (EH) model, wherein the middle portion of one SAM domain (ML surface) binds to the EH surface (c-terminal α5 helix and loop regions) of another SAM domain [99]. An example of the SAM heterotypic interactions through the ML/EH model is evident from the crystal structure of the Anks6-Sam/Anks3-Sam complex. The complex shows that the EH surface of the Anks3 protein interacts with the ML surface of the Anks6 protein [99,100]. On the other hand, neuronal scaffolding protein AIDA-1 contains two tandem SAM domains, which interact homomerically with each other through the ML/EH model (homotypic interaction). In addition to protein–protein interactions, SAM domains are also involved in SAM–lipid and SAM–RNA interactions, wherein they bind with other biological macromolecules with various stoichiometries [95,96,101].

As described in a recent review [99], aberrant SAM–SAM domain interactions may cause pathological cellular responses and conditions. For instance, the SAM domain of TEL mediates its oligomerization with various tyrosine kinases to induce constitutive activation that is essential for cell transformation and with transcription factors that result in the repression of gene transcription [102]. Axon degeneration is an evolutionarily conserved process that clears irreparably damaged axons following nerve injury and also enables developmental axon pruning. Aberrant regulation of axon degeneration leads to dysfunctional neuronal connectivity, as seen in neurodegenerative diseases and neuropathies. The sterile α-motif-containing and armadillo-motif-containing protein 1 (Sarm1) has a SAM domain that forms homotypic interactions. Mutating or deleting the Sarm1–SAM domain abolishes the Sarm1–Sarm1 interaction that is required for axonal degeneration [99,103]. Collectively, these studies suggested that signaling specificity by SAM–SAM domain interactions is mediated by distinct residues on their binding surfaces. The rest of the review will highlight what is currently known about the SAM domain of STIM proteins and their contribution to their physiological functions.

## 5. SAM Domain in STIM Proteins

In addition to the EF-hand domains, STIM proteins have one SAM domain: aa 132–200 in STIM1 and aa 136–204 in STIM2 [104]. In our survey of recent publications and protein databases, we found that STIMs are unique in this regard, as no other proteins contain both EF hand and SAM domains. Similar findings were reached by Stathopulos and colleagues [27]. Human STIM-SAM exhibits five a-helical structures, similar to those reported in other SAM domain-containing proteins [27,28,99]. Internal association of the STIM1 EF-hand hydrophobic cleft with the SAM domain was proposed to keep STIM1 in a compact structure, precluding non-specific interactions and solvent exposure of hydrophobic residues that could destabilize the protein [27]. For instance, to maintain STIM1 in a quiescent state, the C-terminal helix in STIM1-SAM (Leu195, Leu199) is anchored within the hydrophobic cleft of the EF hands (12 residues). Following store depletion, the EF-SAM of STIM1 unravels and oligomerizes to trigger conformational changes in the cytosolic C-terminus and subsequent translocation to the ER-PM junctions. Interestingly, the SAM domain of STIM1 is essential for its activation and puncta formation, as deleting the SAM domain results in continued STIM1 tracking along the microtubule following ER-Ca^2+^ store depletion [105]. In addition, recent studies indicate that amino acid substitutions in the α-7 and α-9 attenuate puncta formation and inhibit SAM multimerization and Orai1 channel activation [106].

Conversely, STIM2 is already partially activated in resting/unstimulated cells, regardless of its higher stability (EF-SAM) than STIM1. The stability of the STIM2-EF-SAM was enhanced by a more extensive hydrophobic cleft (12 residues) in the EF-hands and hydrophobic core (12 residues) in the SAM domain. Additionally, ionic interactions between the Lys103 and Asp200 further stabilize STIM2-EF-SAM [28,104]. Interestingly, while STIM2-EF-SAM is more stable than STIM1-EF-SAM, STIM2-SAM has the lowest oligomerization tendency, while STIM1-SAM shows the greatest tendency. STIM1-EF-SAM exists as a monomer in the presence of Ca^2+^ but forms multimers in its absence. In addition to the stability, EF-SAM of STIM proteins differs in their interhelical angles between cEF-hand (α2 helix) and SAM domain (α10 helix). At the same time, STIM2 adopts a parallel orientation, while STIM1 has a perpendicular orientation. It is interesting to note that STIM from *C. elegans* adopts a similar parallel orientation as STIM2 [107]. The differing cEF-hand-SAM conformations likely influence STIM-SAM domain interactions. Moreover, given the greater STIM2-EF-SAM stability, fractional changes in STIM2 structure may be sufficient to affect its function [108].

An early screen for human SAM (hSAM) domains for polymerization identified several SAM domain-containing proteins in the SMART and NCBI gene databases. A total of 114 unique hSAM domains were identified from 92 proteins in the human genome. Of these, 96 unique hSAM domains from 77 proteins were fused with the super-negatively charged green fluorescent protein (neg-GFP) to analyze their polymerization potential. Interestingly, STIM2 was identified as an hSAM domain that appears to form well-organized polymers instead of simple amorphous aggregates [109]. It is not known why STIM1 was not identified on the same screen, but one possible explanation could be its lower protein expression. STIM proteins were proposed to have evolved to differentially regulate Orai channels through a divergent balance between the Ca^2+^ binding affinity of cEF-hand and SAM domain stability or oligomerization tendency. STIM1-SAM instability and weakly structured Ca^2+^-free EF-SAM enhance STIM1/Orai1 interaction following store depletion. In contrast, STIM2 is more stable and compact under these conditions, which increases the time required for maximal Orai1 channel activation [28]. Combined with the ability of STIM2 cEF-hand to sense smaller changes in [Ca^2+^]_ER_, pre-clustering of STIM2 in ER-PM junctions enables STIM2 to more readily gate Orai1 following cell stimulation. Additionally, STIM2 can also recruit STIM1 into the same ER-PM junctions to enhance Orai1 channel gating, especially with relatively low physiological agonist stimulation [45]. The ability of STIM proteins to form homomers and heteromers enables the cells to tune the response to variable stimuli. However, the role of SAM domains in the STIM2-STIM1 interaction or STIM2 dimerization is yet to be understood.

## 6. Naturally Occurring Mutations in STIM Proteins and Disease

Studies reported over the past several years have revealed significant pathological consequences of mutations in STIM1 that lead to aberrant function. It is important to note that disease-associated mutations in both N- and C-terminal domains have been described only for STIM1 but not for STIM2. A change in the expression level of the STIM2 protein or a modification of the protein that causes functional changes has been reported under certain pathological conditions, and these might contribute to the severity of the condition. Here, we will discuss the mutations in the N-terminal domain of STIM1. Several gain of function (GoF) mutations have been described in the N-termini of STIM1, which cause Tubular aggregate myopathy (TAM) (Figure 3). Of these, five mutations are located in the cEF-hand (Figure 3a), and the remaining is in the nEF-hand [18,110,111,112,113]. H72Q and L96V are located in the E and F helices, respectively, of the cEF-hand. While the STIM1-H72Q mutant exhibits constitutive puncta formation, no clinical symptoms have been reported so far [111]. Although the H72Q mutation does not affect the Ca^2+^-coordinating residues involved in direct Ca^2+^-binding in the cEF loop, it may be involved in stabilizing the intramolecular interactions favoring compactness of STIM1 luminal structure in cooperation with the SAM domain [27,89]. Asn at the 5th position in the cEF loop is important as it directly coordinates Ca^2+^. An Asn to Thr (N80T) mutation has been shown to cause significant STIM1 clustering without store depletion, and the patients exhibit symptoms such as myalgia and high creatinine kinase levels [110]. The other two mutations are at the 6th (G81D) and 9th (D84G) in the cEF loop of STIM1. Studies suggest that both G81D and D84G cause constitutive Ca^2+^ entry and increase in SOCE. In addition, the D84G mutant also exhibits bleeding disorder and malfunctioning of platelets in mice [18,114,115].

Among the nEF-hand mutants, F108I/L, H109N/R/Y, and I115F are located in the loop connecting the two helices at the 1st, 2nd, and 8th positions, respectively (Figure 3b). It is interesting to note that the I115F mutation is seen in patients of both TAM and York platelet syndrome. Mutation of F108 to either Leu (L) or Ile (I) causes proximal muscle weakness and adult-onset myalgia, respectively [110,111]. Sallinger et al. reported that these loop mutants (F108I, H109N/R, and I115F) exhibit high cytosolic Ca^2+^ levels in the absence of stimulus, constitutive activation of STIM1, active Ca^2+^ entry, and increased NFAT nuclear translocation [89,110,111,116]. Biophysical studies such as far-UV CD spectra of F108I indicate a similar α-helical structure as that of wild-type STIM1. However, the F108I mutation destabilizes the protein by affecting its Ca^2+^-binding affinity and overall stability. Of note, STIM1-F108I binds Ca^2+^ with a comparatively weaker affinity and is less thermostable than the wild-type STIM1. Molecular dynamics simulation studies also indicate that the F108I mutation destabilizes the hydrophobic cleft following the unfolding of the nEF hand, whereas the cEF-hand and SAM domain remain stable [89].

In addition to the EF-hand mutations, several studies have reported STIM1-SAM mutations that caused impaired protein function and disease. One study analyzed 37 STIM1 variants found in three pancreatitis patient cohorts, 5 of which are in the SAM domain (E136D, E152K, T153I, Q158E, and T177I). The p.E152K variant is extremely rare in the general population (observed in only three patients) and affects a highly conserved amino acid located in the α2 helix of the SAM domain. HEK293 cells expressing STIM1-E152K showed greater ER-Ca^2+^ store release when stimulated with thapsigargin, tBHQ, and carbachol, resulting in higher Ca^2+^ influx with re-addition of CaCl_2_. The three patients with the *STIM1* p.E152K variant also had heterozygous variants of other chronic pancreatitis susceptibility genes (e.g., *PRSS1* p.P365R, a rare variant). Fibroblasts isolated from a patient with *STIM1* p.E152K and *PRSS1* p.P365R showed enhanced rate and magnitude of Ca^2+^ store release and SOCE. Similar results were obtained when STIM1-E125K was expressed in rat pancreatic AR42J cells following stimulation with cholecystokinin, carbachol, and thapsigargin. AR42J cells expressing STIM1-E152K also showed higher levels of activated trypsin and cell toxicity in unstimulated cells, with further increases following stimulation with cholecystokinin. Mutating the conserved Glu (E152) to Lys (K) destabilizes the Ca^2+^-bound EF-SAM but stabilizes the Ca^2+^-free structure. STIM1-E152K displayed enhanced aggregation due to the destabilized Ca^2+^-bound EF-SAM transitions more readily to form aggregates, and/or the Ca^2+^-free EF-SAM exhibits greater stability following multimerization. The authors proposed that STIM1-E152K interacts with the SERCA pump to modulate its activity and enhance ER re-filling, which may account for the enhanced ER-Ca^2+^ release due to higher [Ca^2+^]_ER_ [117].

STIM1-P165Q is another SAM domain mutation first identified in two siblings who harbor a novel homozygous missense mutation c.C494A in exon 4. PBMCs and PHA-induced T cells obtained from one sibling exhibited decreased SOCE and reduced STIM1 protein expression [118]. Activation was adversely affected by impaired dimer formation of STIM1-P165Q [108]. Another variation within the STIM1-SAM domain, V138I, leads to constitutively higher basal [Ca^2+^]_i_ and enhanced SOCE in HEK293 cells expressing STIM1-V138I. This STIM1-SAM mutant was discovered in a cross-sectional study of patients harboring monoallelic *STIM1* variants and exhibiting musculature disorders [108,119]. STIM1-E136X is a unique mutation where an adenine insertion in exon 3 produced a scrambled sequence of 8 amino acids and a premature stop codon (c.f. wild type STIM1), resulting in undetectable protein levels in affected patients. Interestingly, STIM1-E136X was also identified in Stormorken syndrome patients, who harbor a gain of function mutation in the STIM1 C-terminus. The STIM1-SAM domains also contain several glycosylation sites that control STIM1 function. A double mutant, STIM1-N131D-N171Q, leads to a gain of function effect that allows faster gating of the Orai1 channel. Further studies are required to evaluate how these STIM1-SAM domain mutations affect protein structure and behavior in order to explain their effects on STIM1 function [108].

Unlike STIM1, there is no clear evidence of pathophysiological conditions in humans caused by single nucleotide mutations in STIM2. Although the Clinvar database (https://www.ncbi.nlm.nih.gov/clinvar/?term=STIM2[sym]) (URL (accessed on 5th May 2024) [120] reports several missense variants in 3′UTR and protein-coding regions of STIM2, including the cEF-, nEF-, and SAM domain, the clinical significance of these variants is currently unknown. As these variants have been reported recently, it would be interesting to see future studies evaluating their functional effects and clinical symptoms. Thus, no single nucleotide mutations linked to diseases have been reported for STIM2 so far. Conversely, several studies have elucidated the probable role of STIM2 in pathological conditions related to neurodegenerative diseases, immune function, cancer, etc., which are extensively reviewed in [114,121]. In addition, several studies report abnormal phenotypes in patients having either proximal interstitial deletion [122,123], translocation [124], or duplication [125] in the short arm of chromosome 4 near the STIM2 gene (p15.2). The most common phenotypes observed in those younger patients are physical and facial anomalies, intellectual disability, developmental delay, neurological dysfunction, and congenital heart diseases [114,121,122,123,124,125]. Since multiple genes are involved in those chromosome alterations, it is very difficult to delineate the contribution of STIM2 in those phenotypes. More studies should be focused on understanding the pathological phenotypes associated with the STIM2 gene alterations.

## 7. Conclusions 

STIM1 and STIM2 are unique bi-functional proteins that serve as sensors of [Ca^2+^] in the ER lumen and activators of the Orai Ca^2+^ channels in the plasma membrane. Their structure and regulation allow them to sense changes in ER-[Ca^2+^] and transmit the signal to the plasma membrane channel to trigger Ca^2+^ entry. This mechanism, referred to as store-operated calcium entry, is ubiquitously present in all cell types and involves activation of Ca^2+^ influx into cells via Orai channels in response to agonist-stimulated decreases in ER-[Ca^2+^]. The evolution and distinct functional characteristics speak to the unique function of each protein. STIM2 and STIM1 display high and low Ca^2+^ affinities that permit them to differentially sense minor or major decreases in ER-Ca^2+^, respectively. Further, they also differ in their activation of Orai1, with STIM1 exerting a stronger activation of Ca^2+^ entry than STIM2. These specific properties lead to the tuning of Orai1 activity over a wide range of stimulus intensities. The STIMs have a Ca^2+^-sensing N-terminus, which resides in the ER lumen, a single transmembrane domain that spans the ER membrane, and a C-terminal domain that is located in the cytosol. The N-terminus has a Ca^2+^-binding canonical EF-hand domain, a non-canonical EF-hand domain, and a SAM domain, while the C-terminus has several coiled-coil domains, which include an Orai-binding/activation region (SOAR) and a polybasic tail that binds to plasma membrane lipids. Unbinding of Ca^2+^ from the EF-hand domain is suggested to induce conformational changes in the N-terminus that are then propagated to the C-terminus, causing the SOAR domain to be exposed and the C-terminal poly-basic motif to scaffold to the plasma membrane. Relatively more studies have been carried out with the STIM1 C-terminus than its N-terminus. NMR studies have elaborated the N-terminal structure of STIM proteins, but detailed structure–function analysis is yet lacking. Further studies are required to determine the contributions of non-canonical and SAM domains to the Ca^2+^ sensing and regulation of the protein. Recently, MD-simulation approaches have provided insights into the conformational dynamics of STIM1 N-terminus during Ca^2+^ sensing and activation. However, a lot more data are required to clarify the same for STIM2. In particular, very little is understood regarding the conformational changes in the STIM2-N terminus in response to Ca^2+^. More importantly, there are sparse structural data on the C-terminus of both STIM1 and STIM2. As more mutations related to human diseases are identified for STIM1 and STIM2, it will be necessary to understand the structure–function relationships of the various functional domains of these proteins so that knowledge-based targeting can be utilized for therapies.

## Figures and Tables

**Figure 1 biomolecules-14-01200-f001:**
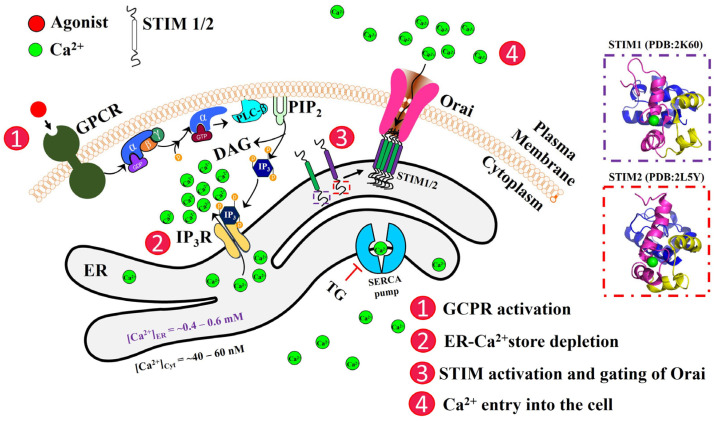
Schematic representation of the activation of store-operated Ca^2+^ entry (SOCE). The cascade of molecular events leading to the ER-Ca^2+^ store depletion and the re-filling up of stores are shown here. STIM proteins are single-pass ER-TM proteins with their N-terminal domains (dashed boxes) within the ER lumen. The NMR structures for the N-terminal EF-SAM of STIM1 (PDB: 2K60 [27]) and STIM2 (PDB: 2L5Y [28]) are shown in dashed boxes. The N-terminus contains a canonical EF-hand (cEF; magenta), a non-canonical EF-hand (nEF; yellow), and a SAM domain (blue). GPCR, G-protein coupled receptor; PLC-ß, phospholipase C-ß; PIP_2_, phosphatidylinositol 4,5-bisphosphate; DAG, diacylglycerol; IP_3_, inositol 1,4,5-trisphosphate; IP_3_R, IP_3_ receptor; ER, endoplasmic reticulum; SERCA, Sarco-ER Ca^2+^; TG, thapsigargin.

**Figure 2 biomolecules-14-01200-f002:**
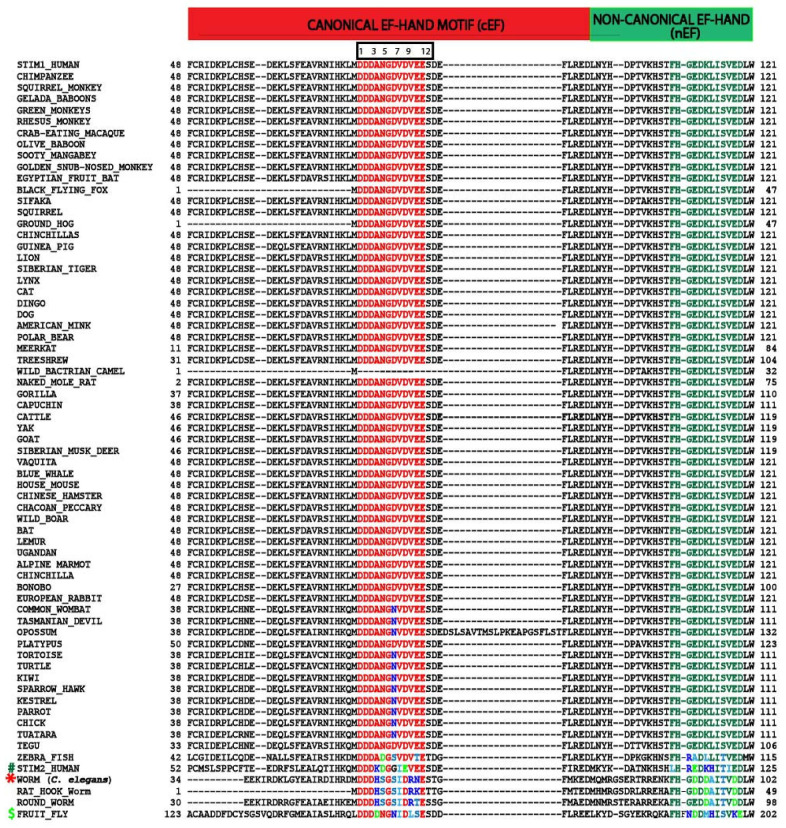
Multiple sequence alignment of STIM1 homologs. Multiple sequence alignment of STIM1 was generated in the MAFFT alignment tool employing BLAST search using human STIM1 (Uniprotkb: Q13586) as a query sequence. The amino acids in the cEF loop and nEF are highly conserved among STIM1 orthologs, shown as red and olive green, respectively. Any variations in the consensus sequence within the loop regions are marked by different colors (dark blue, positively charged amino acids; light green, negatively charged amino acids; light blue, polar uncharged amino acids; turquoise blue, hydrophobic amino acids). Protein sequences of STIM2 (human), STIM from Worm (*C. elegans*), and Fruit fly (*D. melanogaster*) were also included in the alignment, shown as #, *, and $, respectively.

**Figure 3 biomolecules-14-01200-f003:**
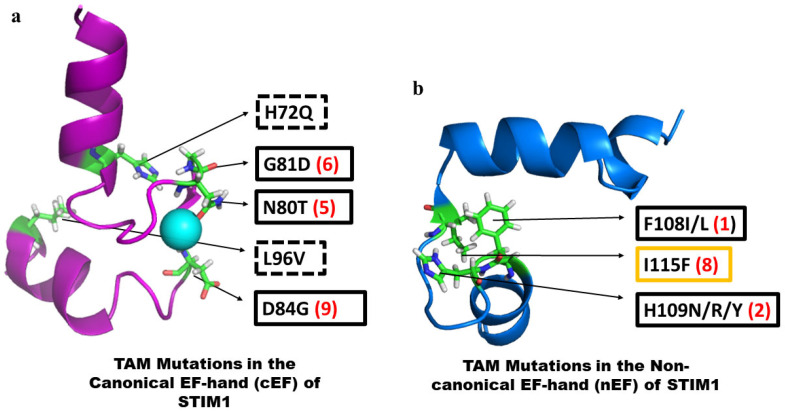
Gain of function (GoF) mutations leading to TAM/York platelet syndrome in canonical (panel (**a**)) and non-canonical (panel (**b**)) EF-hands of STIM1 (PDB: 2K60 [27]). TAM mutations are indicated by either dashed/solid black boxes; the dashed black box around the mutants indicates mutations in the entering/exiting helix. A solid black box indicates mutations in the loop region. The numbers within the paratheses of the mutants represent the position of these amino acids in a typical EF-hand motif; the I115F mutant, seen in both TAM and York platelet syndrome, is shown as an orange box.

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
