# Peer review of "A Deep Dive into the N-Terminus of STIM Proteins: Structure–Function Analysis and Evolutionary Significance of the Functional Domains"

_biomolecules, 2024, doi:10.3390/biom14101200_

Round 1

Reviewer 1 Report

Comments and Suggestions for Authors

This is a very nicely written and interesting review. It provides a thorough review of the structure-function relation of EF-hand motifs and SAM domains of STIM1 and STIM2 proteins. Given the primary role played by these proteins in Ca2+ signaling, this review should be of interest for a large scientific community.

I have a few suggestions of minor changes that the authors could take into account. 

- Lines 37-38: the concept of the target proteins is not very clear. In general, the target is a response. I guess that the authors have in mind secretion processes, or other Ca2+-sensitive physiological responses. Maybe Ca2+-sensitive enzymes could also be mentioned. 

-around line 57. SOCE can be activated when the ER is depleted by other pathways than those involving IP3. For exemple, when they involve RYR.

-line 109. IARC. Please check whether this has been defined.

-lines 381-382. Please indicate references for the KD of STIM1 and STIM2.

-lines 426. remove (???) from the text

Reviewer 2 Report

Comments and Suggestions for Authors

As a part of the special issue of Biomolecules about STIM and Orai, this review focuses on analyzing the N-terminal region of STIM, which contains two EF-hand motifs. The authors begin with some description of the SOCE process, which involves GPCR and PLC in addition to STIM and Orai. For clarity, it should be accompanied with a schematic diagram that offers an overall picture of the process and how the protein molecules as well as the effectors (Ca2+, PIP2, IP3…) are involved by interacting with one another.

The authors then go into more details about Orai and STIM, especially the EF-hand and SAM motifs. Because the 3D structures of Orai and parts of STIM have been known, the authors should also show them as ribbon diagrams in a second figure, so the readers can have some idea about the molecular organization and the location of structural elements like the TM3-TM4 loop in Orai and the EF-hand motifs in STIM. That the C-terminal region of STIM beyond the SAM motif is rich in helices may also be indicated here.

The original Figure 1 should show the residue number of the first aligned amino acid, probably on the left side of each sequence (by replacing the leading dashes).

The words “molecule” in line 64 and “junctions” in line 72 can be removed.

The phrase “similar ionic-sized metal ions” in line 201 needs revision.

Line 426 starts with three question marks. What’s up?

Round 2

Reviewer 2 Report

Comments and Suggestions for Authors

1. Move the word "junction" in line 74 to line 73, and place it after the phrase "ER-plasma membrane".

2. Reconsider the suggestions in the previous comment 2. An illustration of the known structures will allow the reader to have a better view of the significance of this work. The addition of the new Figure 1 is good but it does not show where the EF-hand and SAM are located in STIM and how they may interact with Orai.

---- previous comment 2 ----

The authors then go into more details about Orai and STIM, especially the EF-hand and SAM motifs. Because the 3D structures of Orai and parts of STIM have been known, the authors should also show them as ribbon diagrams in a second figure, so the readers can have some idea about the molecular organization and the location of structural elements like the TM3-TM4 loop in Orai and the EF-hand motifs in STIM. That the C-terminal region of STIM beyond the SAM motif is rich in helices may also be indicated here.
